# A CONSTRUCTIVE PREDICTION OF THE GENERALIZATION ERROR ACROSS SCALES

**Jonathan S. Rosenfeld**[1]  **Amir Rosenfeld**[2]  **Yonatan Belinkov**[13]  **Nir Shavit**[145]
{jonsr,belinkov,shanir}@csail.mit.edu  amir@cse.yorku.ca

[1] Massachusetts Institute of Technology  [2] York University  [3] Harvard University
[4] Neural Magic Inc  [5] Tel Aviv University

## ABSTRACT

The dependency of the generalization error of neural networks on model and dataset size is of critical importance both in practice and for understanding the theory of neural networks. Nevertheless, the functional form of this dependency remains elusive. In this work, we present a functional form which approximates well the generalization error in practice. Capitalizing on the successful concept of model scaling (e.g., width, depth), we are able to simultaneously construct such a form and specify the exact models which can attain it across model/data scales. Our construction follows insights obtained from observations conducted over a range of model/data scales, in various model types and datasets, in vision and language tasks. We show that the form both fits the observations well across scales, and provides accurate predictions from small- to large-scale models and data.

## 1 INTRODUCTION

With the success and heightened adoption of neural networks for real world tasks, some questions remain poorly answered. For a given task and model architecture, how much data would one require to reach a prescribed performance level? How big a model would be needed?

Addressing such questions is made especially difficult by the mounting evidence that large, deep neural networks trained on large-scale data outperform their smaller counterparts, rendering the training of high performance models prohibitively costly. Indeed, in the absence of practical answers to the above questions, surrogate approaches have proven useful. One such common approach is model scaling, where one designs and compares small-scale models, and applies the obtained architectural principles at a larger scale (e.g., Liu et al., 2018; Real et al., 2018; Zoph et al., 2018). Despite these heuristics being widely used to various degrees of success, the relation between the performance of a model in the small- and large-scale settings is not well understood. Hence, exploring the limitations or improving the efficiency of such methods remains subject to trial and error.

In this work we circle back to the fundamental question: *what is the (functional) relation between generalization error and model and dataset sizes?* Critically, we capitalize on the concept of model scaling in its strictest form: we consider the case where there is some given scaling policy that completely defines how to scale up a model from small to large scales. We include in this context all model parameters, such that traversing from one scale (in which all parameters are known) to another requires no additional resources for specifying the model (e.g., architecture search/design).

We empirically explore the behavior of the generalization error over a wide range of datasets and models in vision and language tasks. While the error landscape seems fairly complex at first glance, we observe the emergence of several key characteristics shared across benchmarks and domains. Chief among these characteristics is the emergence of regions where power-law behavior approximates the error well both with respect to data size, when holding model size fixed, and vice versa.

Motivated by these observations, we establish criteria which a function approximating the error landscape should meet. We propose an intuitive candidate for such a function and evaluate its quality, both in explaining the observed error landscapes and in extrapolating from small scale (seen) to large scale (unseen) errors. Critically, our functional approximation of the error depends on both

model and data sizes. We find that this function leads to a high quality fit and extrapolation. For instance, the mean and standard deviation of the relative errors are under 2% when fitting across all scales investigated and under 5% when extrapolating from a slimmed-down model (1/16 of the parameters) on a fraction of the training data (1/8 of the examples) on the ImageNet (Russakovsky et al., 2015) and WikiText-103 (Merity et al., 2016) datasets, with similar results for other datasets.

To the best of our knowledge, this is the first work that provides simultaneously:

- A *joint* functional form of the generalization error landscape—as dependent on both data and model size—with few, interpretable degrees of freedom (section 5).
- Direct and complete specification (via the scaling policy) of the model configuration attaining said generalization error across model and dataset sizes.
- Highly accurate approximation of error measurements across model and data scales via the functional form, evaluated on different models, datasets, and tasks (section 6 ).
- Highly accurate error prediction from small to large model and data (section 7).

We conclude with a discussion of some implications of our findings as a practical and principled tool for understanding network design at small scale and for efficient computation and trade-off design in general. We hope this work also provides a useful empirical leg to stand on and an invitation to search for a theory of generalization error which accounts for our findings.

## 2 RELATED WORK

**Model scaling:**    A number of studies have explored the effect of model scaling on performance. For instance, image classification networks can be scaled by depth (number of layers; He et al., 2016) or width (number of channels; Zagoruyko & Komodakis, 2016; Howard et al., 2017). More recently, Tan & Le (2019) demonstrated how scaling width, depth, and input resolution has combined positive effects larger than scaling each factor in isolation. However, this relationship has yet to be quantified in a predictive form – by how much will error change with model scaling? In this work, we focus on finding a constructive functional form for determining the model given a specified performance.

**Data scaling:**    It has long been recognized that more data improves performance, and various studies report such trends in both computer vision (e.g., Zhu et al., 2012; Sun et al., 2017) and language processing tasks (e.g., Banko & Brill, 2001; Talmor & Berant, 2019). A number of prior studies observed power-law relations between the generalization error and training data size (Cho et al., 2015; Miceli Barone et al., 2017; Johnson et al., 2018). Most relevant to our work, Hestness et al. (2017) explored the effect of data size on the generalization error in vision, language, and speech tasks, and observed a strikingly consistent power-law behavior in a large set of experiments. However, while these studies point to the empirical *existence* of a power law in terms of data, they do not offer tools for predicting the performance given a specified model. Nor do they offer low-cost methods to specify the model configuration which would attain the power law with data dependency. Indeed, Hestness et al. had to search over models and their configurations at large scale to exhibit their findings, incurring prohibitive computational costs.

In contrast, we demonstrate a constructive recipe, where we directly predict the test performance at large scale and specify the full model configuration which attains it (with no need for large-scale search), given performance at small scale.

**Predicting model performance:**    Since training models at full data/model scale may be computationally prohibitive, a line of work tries to predict the performance of a given model on a given dataset, without training the model, for example by using a bank of previously trained models, dataset, and their associated performances (Istrate et al., 2019). Others have proposed to estimate performance on small data (Klein et al., 2017) or model sizes (Zoph et al., 2018; Real et al., 2019) in the context of neural architecture search (NAS). In this case, the small-scale evaluation is used to compare models at small cost, to expedite the search process; see Elsken et al. (2019) for a recent survey. Our work complements previous approaches by demonstrating a functional form that can predict large-scale performance from small-scale measurements. Moreover, our method may be integrated in NAS, addressing some of its current limitations (as discussed in section 8).

Table 1: The datasets and models used in this work, along with their original training data size and the range of explored scales. For more information, see appendix A.

(a) Training data size (number of words) and model size (number of parameters excluding word embeddings) for language modeling tasks.

| Dataset | Size ($N$) | Scales ($n$) | Base Model | Size ($M$) | Scales ($m$) |
|---|---|---|---|---|---|
| PTB | 0.9M | $\left.\begin{array}{l} \\ \\ \end{array}\right\}$ $2^{-k}N$, $0 \le k \le 5$ | AWD-LSTM | 20M | $\left.\begin{array}{l} \\ \\ \end{array}\right\}$ $4^{-k}M$, $0 \le k \le 6$ |
| WikiText-2 | 2M | | AWD-LSTM | 20M | |
| WikiText-103 | 100M | | Transformer-XL | 41M | |

(b) Training data size (number of images) and model size (number of parameters) for image classification tasks.

| Dataset | Size ($N$) | Scales ($n$) | Base Model | Size ($M$) | Scales ($m$) |
|---|---|---|---|---|---|
| ImageNet | 1.2M | $2^{-k}N, 0 \le k \le 6$ | ResNet-50 | 25.5M | $4^{-k}M, 0 \le k \le 6$ |
| CIFAR10 | 60K | $\left.\begin{array}{l} \\ \\ \\ \\ \end{array}\right\}$ $2^{-k}N$, $0 \le k \le 5$ | WRN-44-16 | 0.7M | $4^{-k}M, -3 \le k \le 4$ |
| CIFAR100 | 60K | | WRN-44-16 | 0.7M | $\left.\begin{array}{l} \\ \\ \\ \\ \end{array}\right\}$ $4^{-k}M$, $-2 \le k \le 4$ |
| DTD | 5640 | | WRN-44-16 | 0.7M | |
| Aircraft | 10K | | WRN-44-16 | 0.7M | |
| UCF101 | 13K | | WRN-44-16 | 0.7M | |

**Theoretical error bounds:** Much attention has been given to theoretical explanations of the generalization capabilities of deep neural networks (Neyshabur et al., 2017a;b; Allen-Zhu et al., 2018a;b; Arora et al., 2018). While fully engaging with this literature is beyond our scope, we note that recent studies have derived bounds involving power-law dependencies in both model (Yarotsky, 2018) and data size (Liang et al., 2019). We leave it as an open question for future work to find theoretical explanations for the empirical behavior and the functional form we investigate in this work.

## 3 EXPERIMENTAL SETUP

**Notation:** Let $\mathbb{D}_n = \{\boldsymbol{x}_i, y_i\}_{i=1}^n$ denote a labeled (training) dataset with $n$ samples or datapoints. Let $f_m$ denote a neural network whose size is the number of parameters $m$, such that $\hat{y} = f_m(\boldsymbol{x})$ is the predicted label. Let $\epsilon(n, m)$ be the generalization error as a function of $n$ and $m$, measured by a performance metric (e.g., top-1 accuracy or cross-entropy loss) on a held-out test set. We refer to this error function as the *error landscape*.

### 3.1 SCALING POLICIES

**Dataset scaling:** We wish to scale datasets while preserving the original distribution. For image classification, we uniformly subsample all classes by a constant ratio, thus preserving the relative sample size per class. We limit the maximal sub-sampling to avoid eradicating any class. For language modeling, where the number of classes (vocabulary items) has a very long tail distribution, we randomly sample sentences such that the total number of sampled words will be a certain fraction of the original dataset. Table 1 reports the data scales we use. In all tasks the held-out test set remains untouched for evaluating the error.

**Model scaling:** We are critically interested in a method where moving across scales is defined by some scaling function, such that no additional significant computation would be incurred. We thus consider the case where the model architecture is given and the model size determines how to scale it. For instance, one may scale width (number of channels in convolutional networks, hidden state size in recurrent networks), depth (number of layers), do compound scaling (Tan & Le, 2019), or more generally define a function tying the model degrees of freedom and size. We focus primarily on width scaling in our experiments; the model scales are reported in Table 1. We also perform selected depth scaling to demonstrate flexibility with respect to the scaling method.

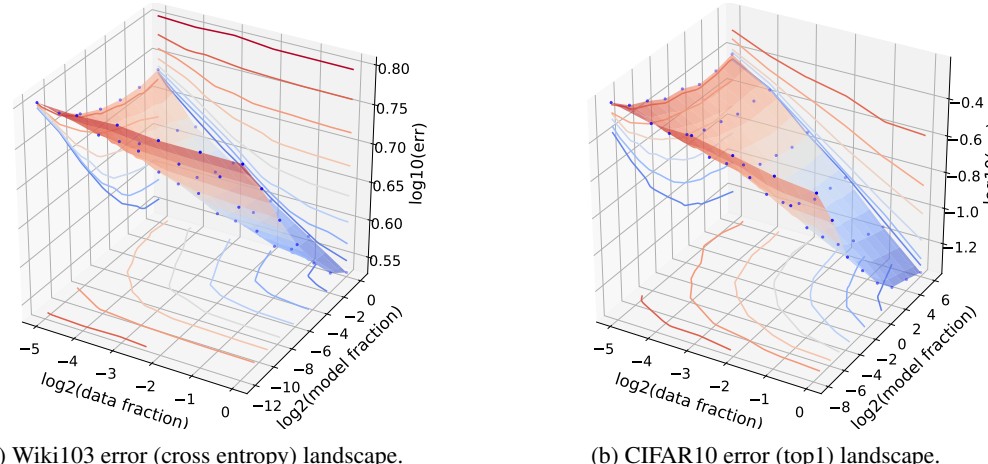

(a) Wiki103 error (cross entropy) landscape.   (b) CIFAR10 error (top1) landscape.

Figure 1: Error landscapes in log-log-log scale. Each point (blue dot) is the error resulting from training with a model/data configuration $m, n$. The surface is a linear interpolation between the points, which is then projected on the $(m, \epsilon)$, $(n, \epsilon)$ and $(m, n)$ planes. See Appendix C for details.

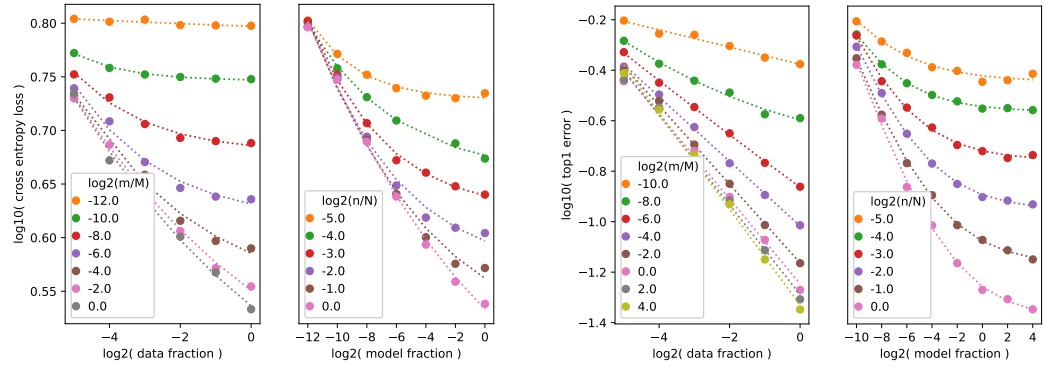

(a) Wiki103 cross entropy vs. data and model size.   (b) CIFAR10 top1 error vs. data and model size.

Figure 2: Error vs. data size (left part of each subfigure) and model size (right part) for Wiki103 and CIFAR10. Solid dots are measurements, dashed lines are best fit to saturating power-law.

**Hyper-parameters:** For similar reasons we wish to avoid hyper-paramater search at large scales, and thus avoid the temptation to tune hyper-parameters accordingly (learning rate, regularization, etc.). Therefore, we hold all hyper-parameters fixed. This enables us to construct a functional form that fits the error landscape and can be used to predict the error across scales while completely defining the model attaining it. We consider pros and cons of this approach in the discussion (section 8).

## 3.2 TASKS, MODELS, OPTIMIZERS AND DATASETS

We experiment with both vision and language tasks. We use 6 benchmark datasets for image classification and 3 for language modeling. For image classification, we train ResNet (He et al., 2016) and WRN models (Zagoruyko & Komodakis, 2016) with stochastic gradient decent (SGD). In section 6.2 we explore the effect of varying architectures and optimizers for a fixed task (CIFAR100), adding VGG16 (Simonyan & Zisserman, 2014) and DenseNet (Huang et al., 2017) models trained with both Adam (Kingma & Ba, 2015) and SGD. For language modeling, we train AWD-LSTM (Merity et al., 2018) and Transformer-XL models (Dai et al., 2019) with SGD and Adam optimizers respectively. Summary statistics are shown in Table 1, along with the range of explored scales. Appendix A gives additional information.

## 4 Observations on the Error Landscape

Figures 1a and 1b respectively show an example test error landscape for width scaling of Transformer-XL on WikiText-103 and WRN-44-16 on CIFAR10. Various additional such landscapes are found in appendix C, showing largely consistent patterns. Examining the error landscapes yields the following observations:

O1 **Model Scaling**

    O1.1 For a given dataset size, scaling up the model results in an initial decrease in test error, which then saturates to a level determined by the dataset size.[1] This behavior has been noted by Tan & Le (2019) across varied model scaling methods, although they have not engaged with the dependency on dataset size.

    O1.2 The rate of error decrease with model size appears well approximated by a power-law.

    These two observations together can be summarized as the following relation:

$$\epsilon(m, n) \approx b(n)m^{-\beta(n)} + c_m(n) \tag{1}$$

    where $b, \beta, c_m$ may depend on the data size $n$, s.t. as $m$ grows, $\epsilon \to c_m$. Example fits to this form (allowing $b, \beta, c_m$ to be fit per $n$) are seen in figure 2a (right) and figure 2b (right).

O2 **Data scaling**

    O2.1 For a given model size, scaling up the dataset results in an initial increase in performance, which then saturates to a level determined by the model size.

    O2.2 The rate of error decrease with dataset size appears well approximated by a power-law. Hestness et al. (2017) also noted a similar relationship, but did not functionally tie the saturation level to the dataset size.

    These two observations together can be summarized as the following relation:

$$\epsilon(m, n) \approx a(m)n^{-\alpha(m)} + c_n(m) \tag{2}$$

    where $a, \alpha, c_n$ may depend on the model size $m$, s.t. as $n$ grows, $\epsilon \to c_n$. Example fits to this form (allowing $a, \alpha, c_n$ to be fit per $m$) are seen in figure 2a (left) and figure 2b (left).

O3 **Joint properties** The behavior of the error when scaling model size while holding data size fixed, and vice versa, extends to the entire error landscape in a well-behaved manner, such that the manifold $\epsilon(m, n)$ is smooth everywhere as a function of both model and data scales.

## 5 Functional Approximation of the Generalization Error

### 5.1 Criteria

Motivated by the above observations, we now consider a functional approximation for the error landscape. In particular, let us consider function families meeting the following criteria which augment and restrict our observations:

    C1 As **either** model or dataset size goes to zero, the expected performance is equivalent to a random-guess error level $\epsilon_0$.[2]

    C2 For a given dataset size, scaling up the model will result in an initial increase in performance, which will then saturate, taking the form in equation 1.

    C3 For a given model size, scaling up the dataset will result in an initial increase in performance, which will then saturate, taking the form in equation 2.

    C4 There exists an irreducible error $\epsilon_\infty$, intrinsic to the dataset.

    C5 The function must be smooth everywhere and monotonic non-increasing in terms of model and data size (observation O3).

While there are many possible function families meeting the above criteria, below we propose a simple function family for our evaluation. We do not claim that this is in fact the true underlying dependency, but rather that it serves as a good approximation of the error landscape—consistent with these criteria.

---

[1]At some point error increase ensues; this point differs between datasets, see Appendix C for examples.

[2]Best guess when $m \to 0$ ($\epsilon_{0n}$) or $n \to 0$ ($\epsilon_{m0}$) need not coincide, but can, e.g., in a balanced dataset.

## 5.2 Proposed Function Family

As a first insightful step, consider the implications of satisfying C2 and C3 *simultaneously*. By examining the limiting behavior as $m$ or $n$ grow, we have:

As $m$ grows large: $\qquad\qquad c_m(n) \approx a(m)n^{-\alpha(m)} + c_n(m)$

As $n$ grows large: $\qquad\qquad c_n(m) \approx b(n)m^{-\beta(n)} + c_m(n)$

Thus, a consistent form satisfying C2 and C3 simultaneously is:

$$\epsilon(m, n) \approx a(m)n^{-\alpha(m)} + b(n)m^{-\beta(n)} + c_\infty \tag{3}$$

where $c_\infty$ is a constant not dependent on either $m$ or $n$.

Let us now examine the simplified case where $a, b, \alpha, \beta$ are constant:

$$\tilde{\epsilon}(m, n) = an^{-\alpha} + bm^{-\beta} + c_\infty \tag{4}$$

where $\alpha \geq 0$ and $\beta \geq 0$ control the *global* rate at which error decreases with data and model size, respectively, $a > 0$ and $b > 0$ are a form of unit conversion between data and model sizes and error, and $c_\infty > 0$ is the asymptotic lower value attainable. This function is a special case of equation 3 and meets criteria C2 and C3 by construction. Importantly C4 and C5 are also met.

However, by giving up the dependence of $a, b, \alpha, \beta$ on $m, n$, this function does not meet criterion C1. We thus need to model the transition from the initial random-guess level to the power-law region. We propose to parameterize the transition using the following envelope (complex) function:

$$\hat{\epsilon}(m, n) = \epsilon_0 \left\| \frac{\tilde{\epsilon}(m, n)}{\tilde{\epsilon}(m, n) - i\eta} \right\| = \epsilon_0 \left\| \frac{an^{-\alpha} + bm^{-\beta} + c_\infty}{an^{-\alpha} + bm^{-\beta} + c_\infty - i\eta} \right\| \tag{5}$$

where $i = \sqrt{-1}$. Here the simple pole at $\eta$ controls the transition point from the initial random-guess level $\epsilon_0$ as $(m, n)$ increase. As $(m, n)$ grow, $\tilde{\epsilon} \to c_\infty$ and the final irreducible error $\epsilon_\infty \triangleq \epsilon_0 c_\infty \eta^{-1}$ is approached. The random-guess error, $\epsilon_0$, is a known parameter determined by dataset statistics (e.g, $(N_{classes}-1)/N_{classes}$ for a balanced dataset). Note that due to our choice of rational envelope, we can divide by a constant the form in equation 4. Without loss of generality, let us choose $a = 1$.

Note that while the forms in equations 3 and 4 are well motivated, the approach taken for modeling the transition is solely a convenience one. In fact, the transition(s) as function of $m$ and $n$ may be captured in the functional forms of $a, b, \alpha, \beta$ or another envelope mechanism. We leave a more refined investigation of the nature of the transitions to future work.

## 6 Error Landscape Estimation

We wish to empirically estimate the quality of the proposed functional parameterization as a fit to the true error landscape. Let $\hat{\epsilon}(n, m; \boldsymbol{\theta})$ be the parametric function family (equation 5) approximating the error landscape $\epsilon(n, m)$, where $\boldsymbol{\theta} = \{\alpha, \beta, b, c_\infty, \eta\}$.[3] Define the divergence $\delta(n, m; \boldsymbol{\theta})$ as the relative difference between the estimated error $\hat{\epsilon}(m, n; \boldsymbol{\theta})$ and the true error $\epsilon(m, n)$:

$$\delta(n, m; \boldsymbol{\theta}) \triangleq \frac{\hat{\epsilon}(m, n; \boldsymbol{\theta}) - \epsilon(m, n)}{\epsilon(m, n)}$$

We fit a least squares regression model to find the best parameters minimizing the divergence. In this section, we fit the function using 10-fold cross-validation across all model/data configurations $m, n$ (see Table 1) and evaluate the fit quality. (In the next section, we perform *extrapolation* experiments, from seen to unseen points.) We perform the fit separately for each dataset and evaluate its quality by the mean $\mu$ and standard deviation $\sigma$ of the divergence $\delta$ over all points $(m, n)$. See Appendix B.1 for experimental details.

As figure 3 shows, estimated test accuracy is highly correlated with actual test accuracy for various datasets, with worst-case values $\mu < 1\%$ and $\sigma < 5\%$ . Note that the number of free parameters is small ($|\boldsymbol{\theta}| \leq 6$) compared to the number of points (42–49 model-data configurations), demonstrating the appropriateness of the proposed function for modeling the complex error landscape.

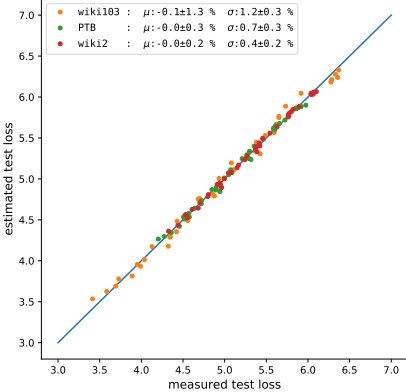

(a) Estimated vs. actual cross-entropy loss for various language modeling datasets.

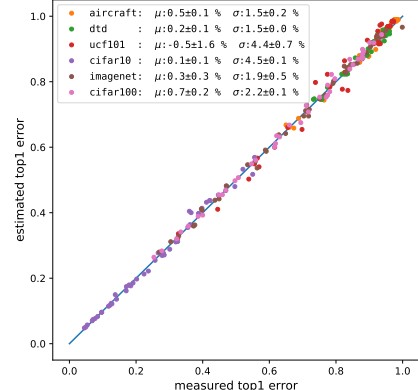

(b) Estimated vs. actual test error for various image classification datasets.

Figure 3: Error estimation results, using 10-fold cross-validation on all configurations in each dataset. For reference, in blue is the identity line. The legend shows mean $\mu$ and standard deviation $\sigma$ of the divergence $\delta$ ($\pm$ one std). See Appendix C for the actual and estimated landscapes in each dataset.

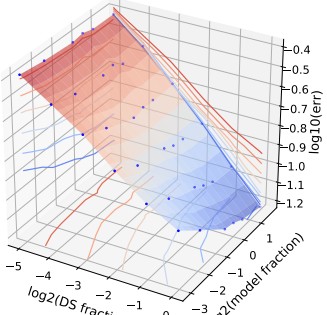

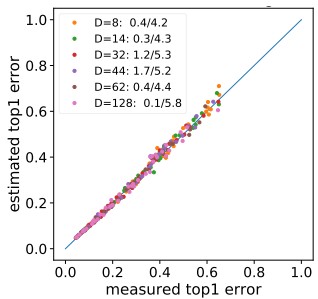

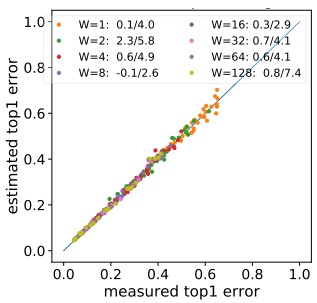

(a) Error landscape when scaling depth (at constant baseline width).

(b) Width scaling fit at different constant depths (D).

(c) Depth scaling fit at different constant widths (W).

Figure 4: Error landscape estimation results on CIFAR10 for width and depth scaling, showing small and comparable fit errors in both cases. Numbers in legends denote mean/variance of the estimation divergence.

## 6.1 A PROBE INTO DEPTH SCALING

Here we verify that our results extend to another canonical scaling policy, namely depth scaling. Figure 4a shows the error landscape with depth scaling on CIFAR10, exhibiting the same characteristics as width scaling. Figures 4b and 4c show error landscape estimation results for both cases of width and depth scaling, exhibiting small and comparable fit errors (confidence intervals $< 3\%$).

Since the difference in approximation quality is effectively indistinguishable when scaling depth or width orthogonally, we expect compound scaling to adhere to the same functional form. Indeed, we verified this on the publicly available (model scaling only) results for EfficientNet (Tan & Le, 2019).

## 6.2 ON THE VARIETY OF OPTIMIZERS AND ARCHITECTURES

Our study covers a deliberate variety of architectures (ResNet, WRN, LSTM, Transformer) and optimizers (Adam, SGD variants), following standard implementations in the literature as recommended for each dataset/model setting; see Appendix A.

---

[3] For image classification, we set $\epsilon_0 = (N_{classes} - 1)/N_{classes}$ (the balanced dataset case). For language modeling, we estimate $\epsilon_0$ as another parameter, such that $\boldsymbol{\theta} = \{\alpha, \beta, b, c_\infty, \eta, \epsilon_0\}$ in this case.

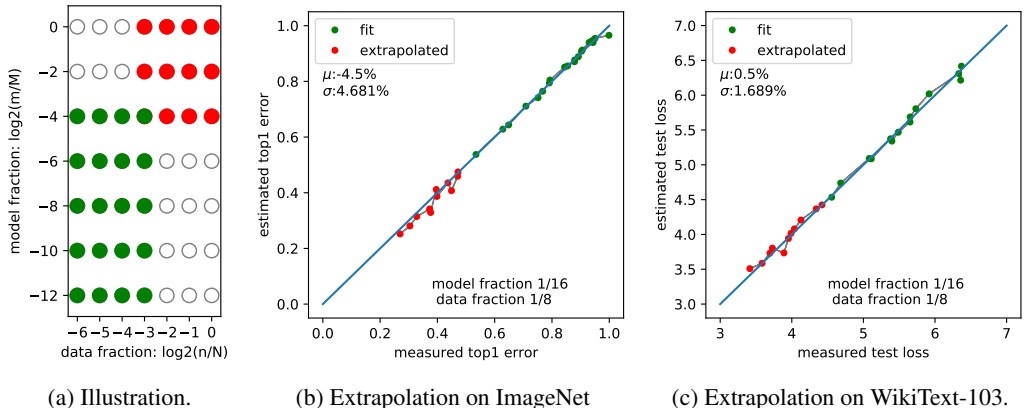

(a) Illustration.   (b) Extrapolation on ImageNet   (c) Extrapolation on WikiText-103.

Figure 6: Extrapolation results. (a) Illustration of the extrapolation setup, where we fit on a subset of the points (in green) and predict on larger points (in red). (b) and (c) show example results on one configuration in two benchmark datasets. Comprehensive results are given in Appendix D.

However, the model/optimizer settings differ in multiple aspects across the different tasks , rendering the comparison of, say, different optimizers, challenging. In this section we verify that the functional form holds when varying the optimizer and/or the architecture on the same task, namely image classification on CIFAR100.

In addition to the previously examined setting of WRN with SGD, we add four more settings: two well known architectures (VGG and DenseNet), each trained with both SGD and Adam optimizers. See Appendix A for experimental details. Figure 5 exhibits consistent, accurate, fit values across all architecture/optimizer settings, with mean divergence of $\mu < 1\%$ (std: $\sigma < 6\%$; confidence intervals $< 4\%$).

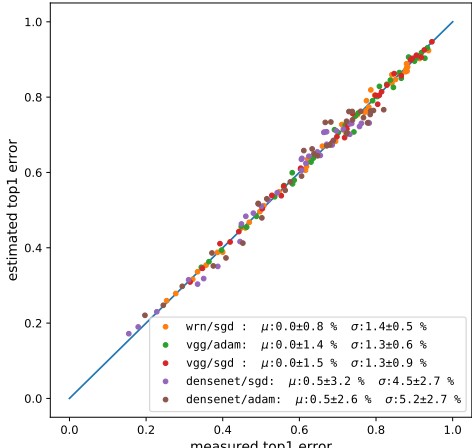

Figure 5: CIFAR100 Error estimation results with three architectures (WRN, VGG, DenseNet) and two optimizers (SGD, Adam).

## 7 EXTRAPOLATION

In this section, we evaluate the ability of our functional approximation to extrapolate beyond seen model/data configurations. The primary question we ask is: can we predict the error of a large model/data configuration from the errors of smaller-scale model/data configurations? To do this, we fit the least squares regression on a subset of the configurations and predict the error on larger, unseen configurations. More formally, let $(m_i, n_j)$ denote a given model/data configuration. We first estimate parameters $\boldsymbol{\theta}_{ij}$ by fitting the function in equation 5 on all points of at most that size ($m \leq m_i, n \leq n_j$). Then we predict the error $\epsilon(m, n)$ in all points corresponding to larger configurations ($m > m_i, n > n_j$) using estimated $\boldsymbol{\theta}_{ij}$. Finally, we measure the divergence $\delta(m, n)$ between the estimated error and the actual error at all larger configurations. This process is illustrated in figure 6a.

Figure 6b shows the results of one such extrapolation experiment, on ImageNet. In this case, we have fit the functional form on all configurations of model size $m \leq m_i = M/16$ and data size $n \leq n_j = N/8$, and predicted the error on all larger configurations. As the figure shows, the extrapolation is highly accurate, with a mean divergence of $\mu = 4.5\%$ (std: $\sigma = 4.7\%$). Figure 6c reports a similar experiment on WikiText-103. Here, again, we see very good extrapolation, with a mean divergence of $\mu = 0.5\%$ (std: $\sigma = 1.7\%$). Note that each extrapolation is run 10 times with different random initializations of $\boldsymbol{\theta}_{ij}$ in the least squares with negligible effect on the prediction.

In practice, we may be interested in extrapolation quality with different subsets of configurations. Appendix D provides detailed extrapolation results on multiple subsets of configurations, for both vision and language datasets. Generally, the extrapolation performs well once not ill-posed, which may be caused by lack of signal in the region of the initial "random-guess" level, or in degenerate cases like having fewer measurements than the number of free parameters in $\boldsymbol{\theta}$.

## 8   DISCUSSION AND CONCLUSION

In this work, through insights gained by the joint examination of the dependencies of generalization error on both model and data size, we arrive at criteria for functions consistent with the form of the generalization error under a given scaling policy. We consider one such function and find it to be in very good agreement with the actual behavior of the error landscape. Indeed, the agreement is strong enough that *extrapolation* from small to large scale becomes feasible: the function predicts the behavior of the generalization error in practice for the practical case of scaling models and data. We discuss several example implications of knowing such a functional form.

**Small-scale network development:**   At the core of small fidelity searches is the notion of *performance rank* comparison between models. However, small scale and large scale ranks are not assured to be consistent. If indeed a functional form such as empirically found in this work holds very generally, then in contrast, one can safely assess *scaling rank* between models at small scale, with the assurance that it remains consistent. This suggests that one would be well served by searching over scaling policies; a pertinent example of such a success is Tan & Le (2019). The functional form also explains the limitation of small-scale search: once reaching the random-guess error level, where the sensitivity to scaling vanishes, the informativeness of ranking diminishes. Finally, the functional form allows direct usage of differentiable methods for NAS.

**Principled design:**   Knowing the error landscape function facilitates reasoning about the choice of $(m, n)$ attaining a specified error level. In other words, for any given error level, one can solve Eq. 5 for $m, n$ based on small-scale measurements. Thus, one can quantitatively answer design questions regarding the expected (in particular, large-scale) relations between $m$, $n$, and $\epsilon$. In fact, Eq. 5 provides direct ansewrs to questions such as "how much data would one require to reach a prescribed performance level?" or "how big a model would be needed?" Imposing constraints is also straightforward. For instance, consider the following question: "What is the maximal model size possibly needed (useful), when the data is limited in size, $n = n_{lim}$ (for a given model architecture and scaling policy)?" For a fixed dataset size, model scaling eventually contributes marginally to error reduction and becomes negligible when $bm^{-\beta} \ll n_{lim}^{-\alpha}$ (Eq. 5). Define the relative contribution threshold $T$ as satisfying $T = \frac{n_{lim}^{-\alpha}}{bm_{max}^{-\beta}}$. (For example, $T = 10$.) Then the maximal useful model size meeting threshold $T$ is:

$$m_{max}(T) = (bT)^{1/\beta}\, n_{lim}^{\alpha/\beta}$$

Similarly, The maximal useful amount of data for a limited sized model $m_{lim}$ is:

$$n_{max}(T) = (1/bT)^{1/\alpha}\, m_{lim}^{\beta/\alpha}$$

Moreover, Eq. 5 allows for complex design trade-offs. Generally, given some design-tradeoff cost function $C(m, n, \epsilon)$, one can minimize such cost s.t. Eq. 5. For example, consider the case of optimizing for efficient computation which has both practical and environmental importance (Schwartz et al., 2019). Since the number of FLOPs during training is $\propto m \cdot n$ (for constant epoch budget), the trade-off cost function may be formulated as $C(\text{FLOPS}, \epsilon) = C(mn, \epsilon)$. Further, since constant error contour is very well approximated by $c = \frac{1}{n^\alpha} + \frac{b}{m^\beta}$ (Eq. 5), dataset and models may be scaled with optimal resource efficiency with no effect on performance by solving for:

$$\underset{m,n}{\operatorname{argmin}} \quad m \cdot n \qquad \text{s.t.} \quad c = \frac{1}{n^\alpha} + \frac{b}{m^\beta}$$

The solution gives us the optimal-computational-efficiency ratio of model to data size: $\frac{b\beta}{\alpha}\frac{n^\alpha}{m^\beta} = 1$.

**Limitations:** We have made a few simplifying assumptions in our choice of approximating function, in particular in how to model the transition from the initial random-guess error level and the union of the random-guess level of the two scenarios (small model with large data and large model with small data). We leave a more detailed examination of the behavior of the transitions from random-guess error levels and refinements of the functional form to future work.

Critically, the restrictive nature of our scaling framework (all parameters and hyperparameters described by a policy) is both a blessing and a challenge. The blessing comes in fulfilling the goal of finding simultaneously both the form of the generalization error and the full specification of the model and hyperparameters that attain it across scales. The challenge is that we have demonstrated in this work only the case of constant hyper-parameters. We conjecture that the relation between model configuration and hyperparameter choice (Zela et al., 2018) may entail the potential to formulate hyperparameter-scaling policies similar in nature to the model-scaling polices, and that these too fall under the scope of the form we find in this work. This too will be the subject of future work.

We hope that this work will bring the actual functional form of the generalization error in this practical case of scaling to the fore, both in practice and as an empirical leg to stand on in the quest for its theoretical origins.

ACKNOWLEDGMENTS

We thank Alexander Rakhlin, Alexander Madry, Kai Xiao, Lu Mi, Viaks Garg, Dan Alistrah, and Tommi Jaakkola for discussions and their help. We also thank the anonymous reviewers for their valuable feedback. J.R. was partly supported by the Eli and Dorothy Berman Fellowship as well as grants NSF IIS-1447786, NSF CCF-1563880 and China-Singapore Suzhou Industrial Park. A.R. was partially supported by the Air Force Office of Scientific Research USA (FA9550-18-1-0054) though a grant to John K. Tsotsos. Y.B. was partly supported by the Harvard Mind ,Brain, and Behavior Initiative.

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

# A   Datasets and Models

## A.1   Image Classification

### A.1.1   Datasets

We evaluated our predictions on several popular image classification datasets: ImageNet (Russakovsky et al., 2015): a large-scale recognition benchmark consisting of natural images of 1000 object categories with 1.28M training images spread roughly uniformly over the categories. It has 50K validation and 100K testing images. It has been the most popular large-scale benchmark for image classification methods for the better part of the last decade. CIFAR10/100 (Krizhevsky et al., 2009): 60K natural RGB images of 10 classes (100 for CIFAR100) with a train/test split of 50K/10K. For each of the following datasets, we use the version collated, resized, and split into train/validation/test sets by Rebuffi et al. (2017). DTD (Cimpoi et al., 2014): a texture database of 47 categories and 5640 images. Aircraft (Maji et al., 2013): 10K images of 100 different aircraft classes. UCF101 (Soomro et al., 2012): originally a video action recognition dataset, converted using the method of Bilen et al. (2016) into a single image per video. It contains 13,320 images of 101 action classes.

### A.1.2   Models

We experiment with four models for image classification. We use different variants of the popular ResNet architecture (He et al., 2016) in the main experiments. For ImageNet we use ResNet-50 and build on the code from the PyTorch framework (Paszke et al., 2017) to vary the model width. For all other datasets we use WRN-44-16 (Wu et al., 2016) of varying widths, modified from the implementation of Hoffer et al. (2018).

Scaling the models' width is performed by multiplying the number of channels in each convolutional layer and the width of the hidden linear layers by a constant factor and rounding to the nearest integer. The ranges of width scales (and data scales) for the main experiments are detailed in Table 1b.

In section 6.2, we perform width scaling for two additional architectures, VGG16bn (Simonyan & Zisserman, 2014) and DenseNet (L=40, k=32) (Huang et al., 2017). The VGG and DenseNet models were also modified for width scaling from the implementation of Hoffer et al. (2018). The model scales in this case are $4^{-k}$, $0 \le k \le 5$, for both VGG and DenseNEt.

Depth-scaling, in the CIFAR10 case (section 6.1), is performed by appending extra layers within each block.

### A.1.3   Training

In the main experiments, training is done via SGD with a momentum of 0.9, weight decay of 1e-4 and initial learning rate of 0.1. For ImageNet we train for 90 epochs, decreasing the learning rate by a multiplicative factor of 0.1 after and 30 and after 60 epochs. We use a batch size of 16. For all other vision datasets we use a batch-size of 128. We begin training with a learning rate of 0.1, run for 200 epochs, and reduce by a multiplicative factor of 0.1 after 80, 120, and 160 epochs.

For the VGG and DenseNet experiments on CIFAR100 in section 6.2, we train with both SGD and Adam optimizers. We train VGG for 170 epochs and Densenet for 300 epochs. Adam hyperparameters are default, with an initial learning rate of 1e-3. When training with SGD, we retain initial learning rate, batch size, momentum, and weight-decay, as in the main experiment (at 0.1, 128, 0.9, and 1e-4 respectively) and follow standard stepped learning rate schedules: For VGG, learning rate multiplicative factor of 0.1 after 80, 120, and 160 epochs; For DenseNet, learning rate multiplicative factor of 0.1 after 150 and 225 epochs.

## A.2   Language Modeling

### A.2.1   Datasets

We evaluate on several datasets commonly used for (word-level) language modeling: Penn Treebank (Mikolov et al., 2010), WikiText-2 (Bradbury et al., 2017), and WikiText-103 (Merity et al., 2016). The PTB is a relatively small language modeling dataset of news texts, with a vocabu-

lary of 10K unique words and about 900K/70K/80K training/validation/test words. WikiText-2 is drawn from Wikipedia articles and it is both larger and richer, with a vocabulary of 33K words and 2M/210K/240K training/validation/test words. WikiText-103 is also based on Wikipedia, but larger still, with a vocabulary of 270K words and 100M training words (and the same validation and test sets as WikiText-2).

### A.2.2 MODELS

We experiment with two standard models for language modeling: Transformer-XL (Dai et al., 2019) and AWD-LSTM (Merity et al., 2018). Transformer-XL is a recent language modeling architecture that is based on transformer self-attention (Vaswani et al., 2017), but modified to better learn dependencies beyond a fixed length by adding a segment-level recurrence mechanism. It has achieved state-of-the-art results on multiple benchmarks. We use the official PyTorch implementation[4] with their base configuration: 16 layers, embedding size of 410, inner dimension of 2100 in the fully-connected layers, and 10 attention heads. Training is done with Adam. See the implementation for other details. For scaling experiments, we decimate the inner dimension. We use Transformer-XL for WikiText-103.

AWD-LSTM is a long short-term memory (Hochreiter & Schmidhuber, 1997) language model with adaptive weight averaging. We use the official implementation[5] with the recommended configuration: 3 layers, embedding size of 400, and hidden state size of 1150. Training is done with SGD. We use AWD-LSTM for PTB and WikiText-2 and follow the recommended settings for these two datasets. For scaling experiments, we decimate the hidden state size.

---

[4]`https://github.com/kimiyoung/transformer-xl`
[5]`https://github.com/salesforce/awd-lstm-lm`

# B    ERROR ESTIMATION EXPERIMENT

## B.1    EXPERIMENTAL DETAILS

In the experiment described in section 6, we fit a least squares regression model to find the best parameters minimizing the divergence $\delta(m, n)$ - evaluated at configurations $m, n$ as in Table 1:

$$\boldsymbol{\theta}^* = \arg\min_{\boldsymbol{\theta}} \sum_{n,m} |\delta(m, n; \boldsymbol{\theta})|^2$$

We quantify the quality of the fit by the mean $\mu$ and standard deviation $\sigma$ of the fitted divergence by performing standard 10-fold cross validation over all points $(m, n)$ with confidence intervals reported as $\pm 1$ std over the folds.

## B.2    FOUND THETA VALUES

Table 2: Optimal values of $\boldsymbol{\theta}$ as found by the least squres regression fitting the functional form.

(a) Image classification (fitting top 1 error).

|  | $\alpha$ | $\beta$ | $b$ | $c_\infty$ | $\eta$ |
|---|---|---|---|---|---|
| ImageNet | 0.75 | 0.61 | 0.76 | 3.63 | 18.50 |
| CIFAR10 | 0.66 | 0.53 | $5.87 \cdot 10^{-02}$ | $7.14 \cdot 10^{-14}$ | 19.77 |
| CIFAR100 | 0.70 | 0.51 | 0.15 | 0.71 | 6.93 |
| DTD | 0.40 | 1.16 | $4.30 \cdot 10^{-05}$ | $1.27 \cdot 10^{-09}$ | 0.85 |
| Aircraft | 1.10 | 0.83 | $3.47 \cdot 10^{-03}$ | $5.16 \cdot 10^{-10}$ | 1.13 |
| UFC101 | 0.93 | 0.54 | $4.68 \cdot 10^{-02}$ | $1.16 \cdot 10^{-09}$ | 2.98 |

(b) Language modeling (fitting cross entropy loss).

|  | $\alpha$ | $\beta$ | $b$ | $c_\infty$ | $\eta$ | $\epsilon_0$ |
|---|---|---|---|---|---|---|
| PTB | 0.81 | 0.34 | 0.15 | 5.00 | 6.27 | 6.10 |
| WikiText-2 | 1.01 | 0.22 | 0.99 | 8.23 | 10.38 | 6.21 |
| WikiText-103 | 0.74 | 0.56 | 0.33 | 9.04 | 16.34 | 6.60 |

## C    ADDITIONAL ERROR LANDSCAPE MEASUREMENTS AND ESTIMATIONS

In this appendix, we provide error landscape measurements and estimations for all datasets, corresponding to the experiment in section 6. The results are shown in 3D graphs similar to figure 1. In each such graph, the z-axis is the logarithm of the generalization error as a function of two independent variables: the model size $m$ and the data size $n$.

The 3D graph is deliberately portrayed in log-log-log scale, as we cover a very large range of data scales and model scales and a correspondingly wide range of errors. This view is a useful one when one wishes to evaluate both large dynamic ranges (simultaneously both very large and very small values) and is especially vivid in portraying power-law like dependencies; a power-law naturally forms a straight line in a log-log view.

In each figure, subfigure (a) shows the measured error landscape is in log-log-log scale, where each point (blue dot) is the error resulting from training with a model/data configuration $m, n$. Subfigure (b) shows the best-fit estimated error landscape. The surface is a linear interpolation between the points, which is then projected on the model-error $(m, \epsilon)$, data-error $(n, \epsilon)$, and model-data $(m, n)$ planes. The contour plots on each one of these planes are the projections of the error landscape surface, and are useful in considering the behavior of the surface when holding one dimension constant.

We call to attention several interesting observations on the datasets explored:

- As quantified rigorously in section 6, the fits perform well across error ranges. In these surfaces, one also gets qualitative sense of the fit adequacy across the wide ranges of the dataset and model scales directly. While perhaps slightly difficult to asses the surface directly, a helpful view is to consider the similarity between the projections of the actual and projected surfaces.

- With increasing model size, indeed typically the error does remain saturated. However, in one of our tested datasets (figure 12) there was a renewed slight increase. We verify that this is indeed over-fitting, in the sense that there is no corresponding increase in the *training* error. We note that the functional form we find can actually be used to veer clear of the $m, n$ regions where such over-fitting may occur.

- The simplifying approach taken by considering the random guess levels (and associated transitions) for small models or small data as identical, seems to work fairly well with some deviation apparent by examining figure 15. Indeed the simplification can hold well for balanced datasets, but need not for imbalanced ones such as in the task of language modeling. Thus, a relaxation of this simplification is expected to be important conceptually and practically.

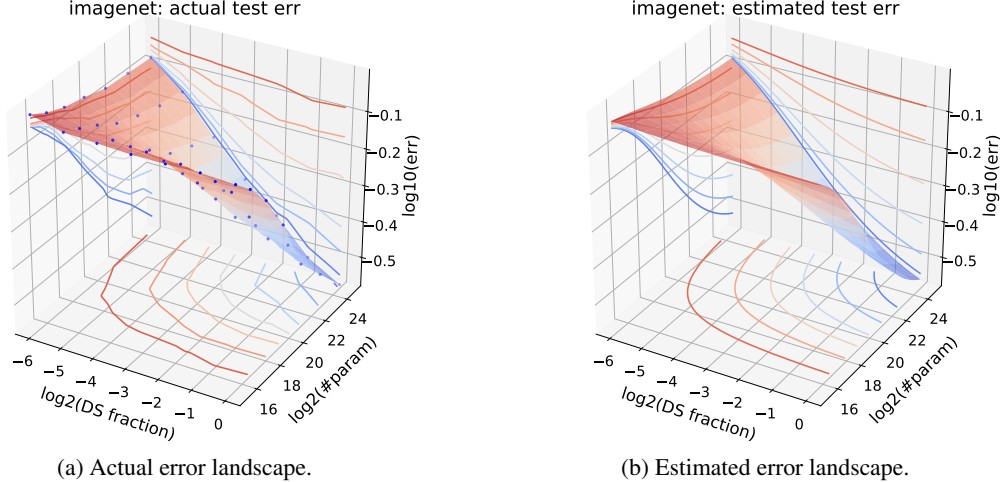

(a) Actual error landscape.

(b) Estimated error landscape.

Figure 7: ImageNet error landscape.

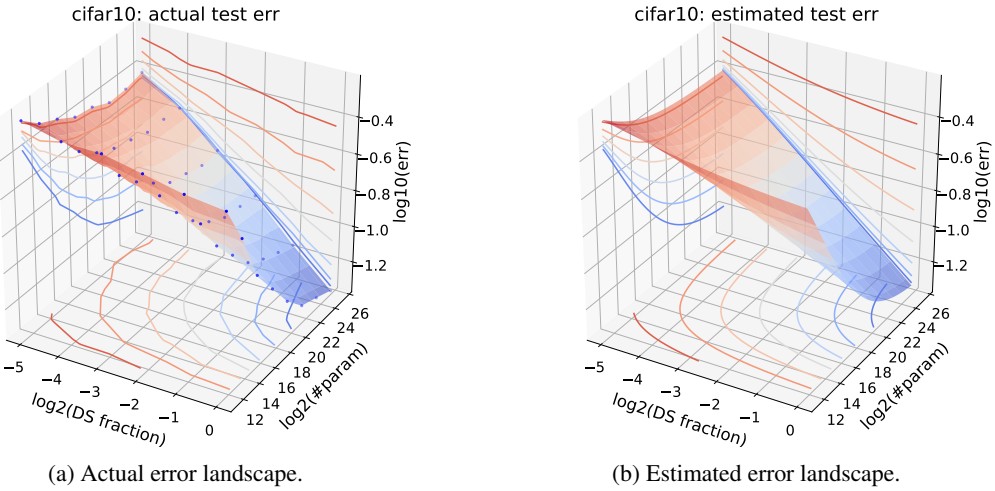

(a) Actual error landscape.

(b) Estimated error landscape.

Figure 8: CIFAR10 error landscape.

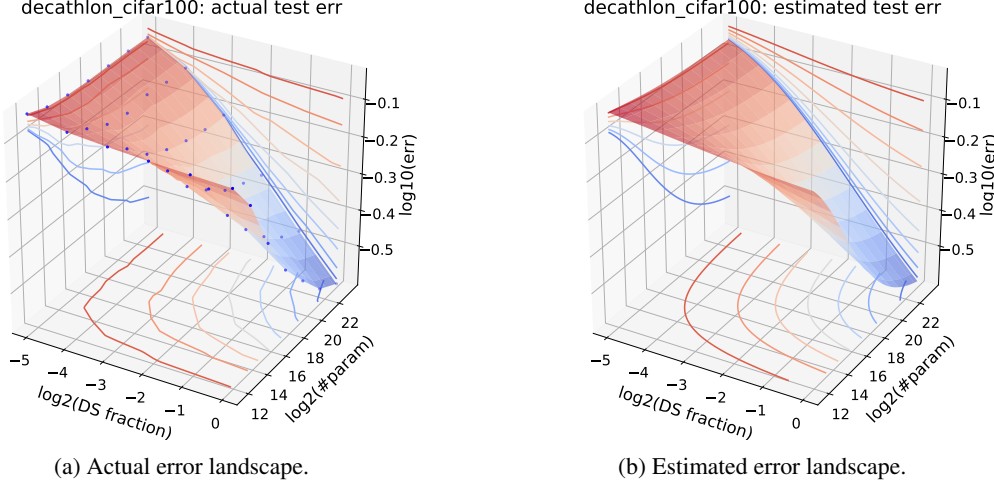

(a) Actual error landscape.

(b) Estimated error landscape.

Figure 9: CIFAR100 error landscape.

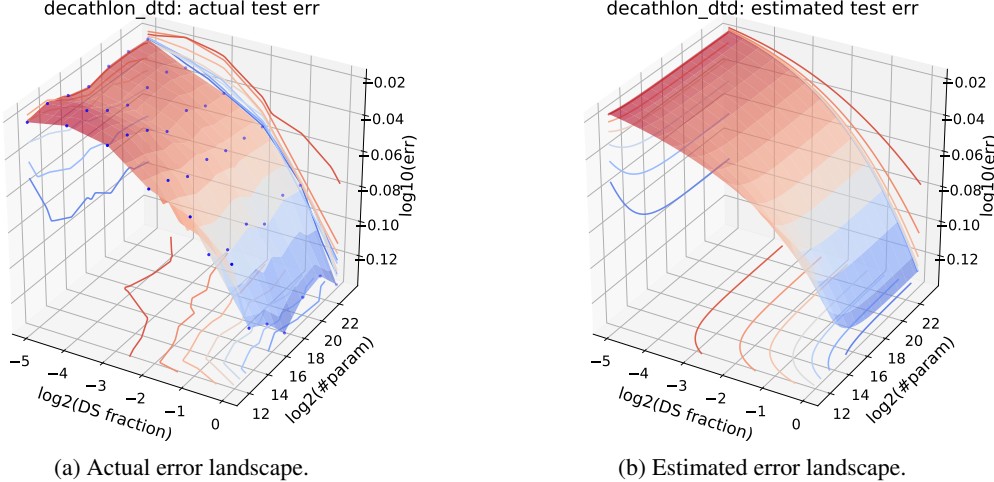

(a) Actual error landscape.

(b) Estimated error landscape.

Figure 10: DTD error landscape.

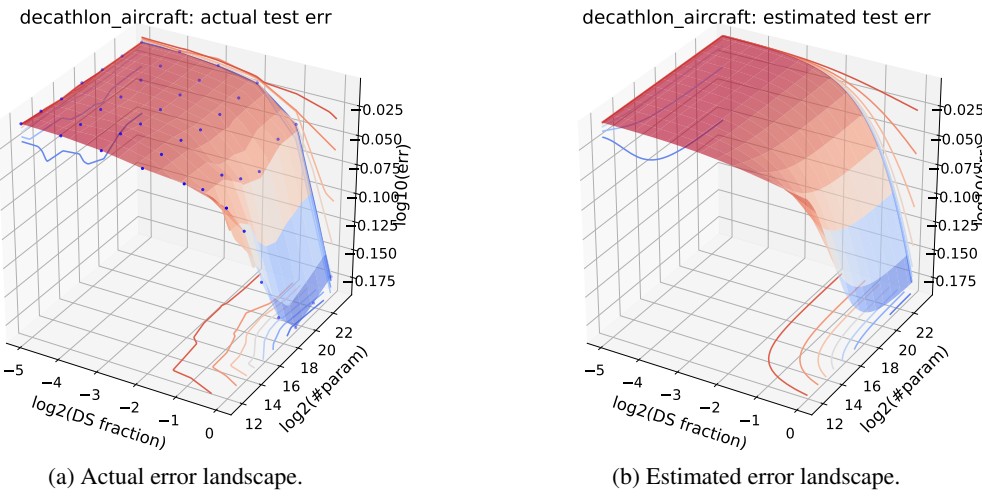

(a) Actual error landscape.

(b) Estimated error landscape.

Figure 11: Aircraft error landscape.

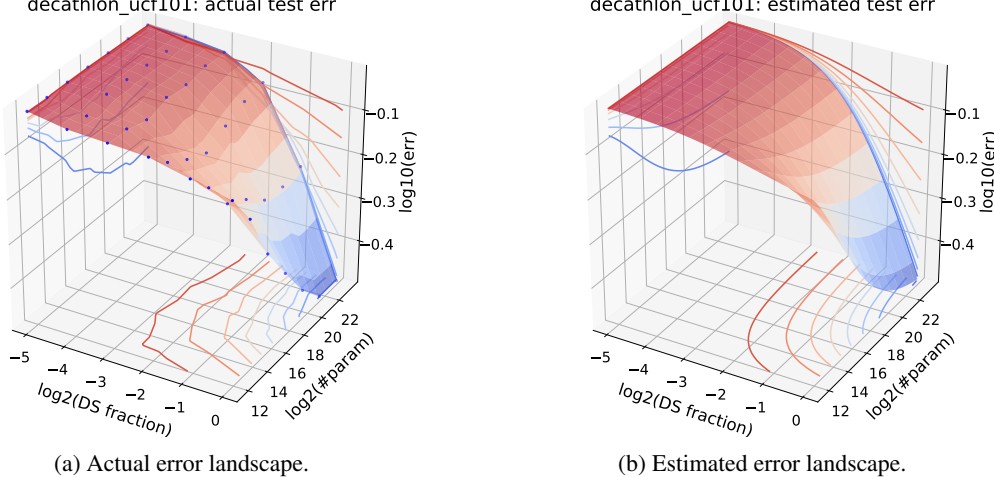

(a) Actual error landscape.

(b) Estimated error landscape.

Figure 12: UFC101 error landscape.

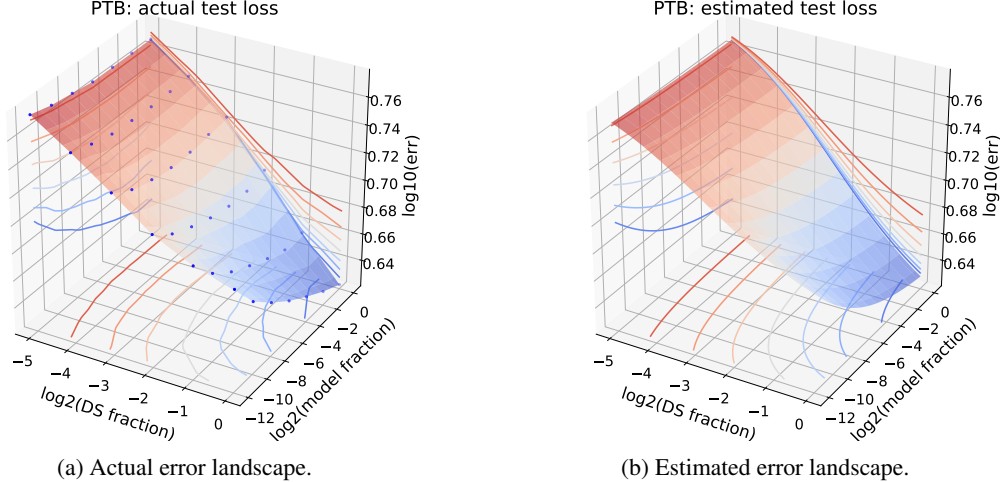

(a) Actual error landscape.

(b) Estimated error landscape.

Figure 13: PTB error landscape.

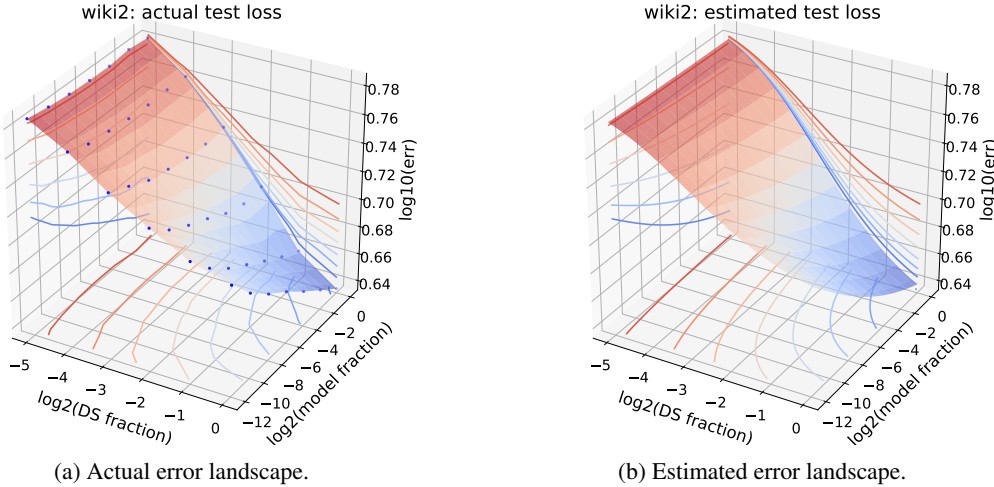

(a) Actual error landscape.

(b) Estimated error landscape.

Figure 14: WikiText-2 error landscape.

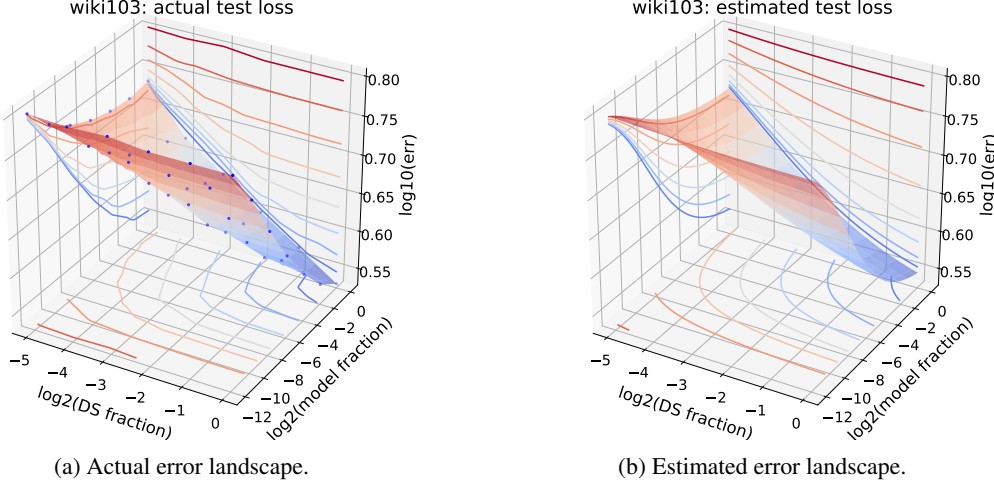

(a) Actual error landscape.

(b) Estimated error landscape.

Figure 15: WikiText-103 error landscape.

## D    ADDITIONAL EXTRAPOLATION RESULTS

Here we provide detailed extrapolation results, for all datasets. All figures are structured in a similar way. Each subplot shows estimated (y-axis) vs. actual error (x-axis) (0 to 1 scale on both axes). Each subplot is located at the coordinate of the maximal data and model given for the task of performing the fit to the functional form in equation 5. This is the point at the top-right corner of the green dots in the illustration in figure 6a. The target is to find the error-landscape values for unseen, larger scales of both model and data (red points in the same illustration). Going from left to right in each figure indicates observed measurements of the error from models of an increasing fraction w.r.t the full size. Going from bottom-to top indicates observed measurements of the error from dataset sizes of an increasingly large fraction of the full dataset.

In each subplot, every point shows the estimated vs. actual error on a model-data configuration. Points that were given for fitting the function are colored in green, while unseen points that were not used are in red. The red points show the estimation error vs. actual error when extrapolating to all larger models and data sizes. In each subplot, the mean and standard deviation over all divergences $\delta$ at target points are given in text.

Each experiment fit of the parameters was repeated 100 times, with different random initializations of $\boldsymbol{\theta}$. The shaded bands show one standard deviation across these runs.

The quality of the extrapolation is critically dependent on the signal provided in the (green) fitted points. Two limiting factors are evident by examining the figures below, which both play a role in the well-posedness of the solution:

- The proximity to the initial random guess level. Only upon transitioning from the initial error plateau, does meaningful signal about the scaling rates become available. Indeed, for scales prior still in the region or close to the initial error level, one sees poor extrapolation results; see figures 18, 19, and 21, and the vivid origin of this phenomena by examining figures 11, 10, and 12.

- A second source of ill-posedness is tied to the number of configurations used for the estimation of $\boldsymbol{\theta}$. Clearly, when this is small, one cannot expect the extrapolation to be stable. In fact, at least two measurements in each scaling dimension (model/data) are needed, and no less than the number of parameters in $\boldsymbol{\theta}$ in total. Indeed, for all the plots in this appendix, the smallest scale of $m, n$ is omitted form the graph such that the lowermost row and leftmost column span exactly two model and data scales correspondingly. Of course, there is nothing tying directly the number of points and scale of configurations measured, and one can decouple these two factors by taking closer spaced samples at small scale.

- When both the above factors are not limiting the measurement, one readily sees that for divergences of no more than a few percent, it is sufficient to measure model/data configurations which are far-ranged from the configurations which one wishes to extrapolate to .

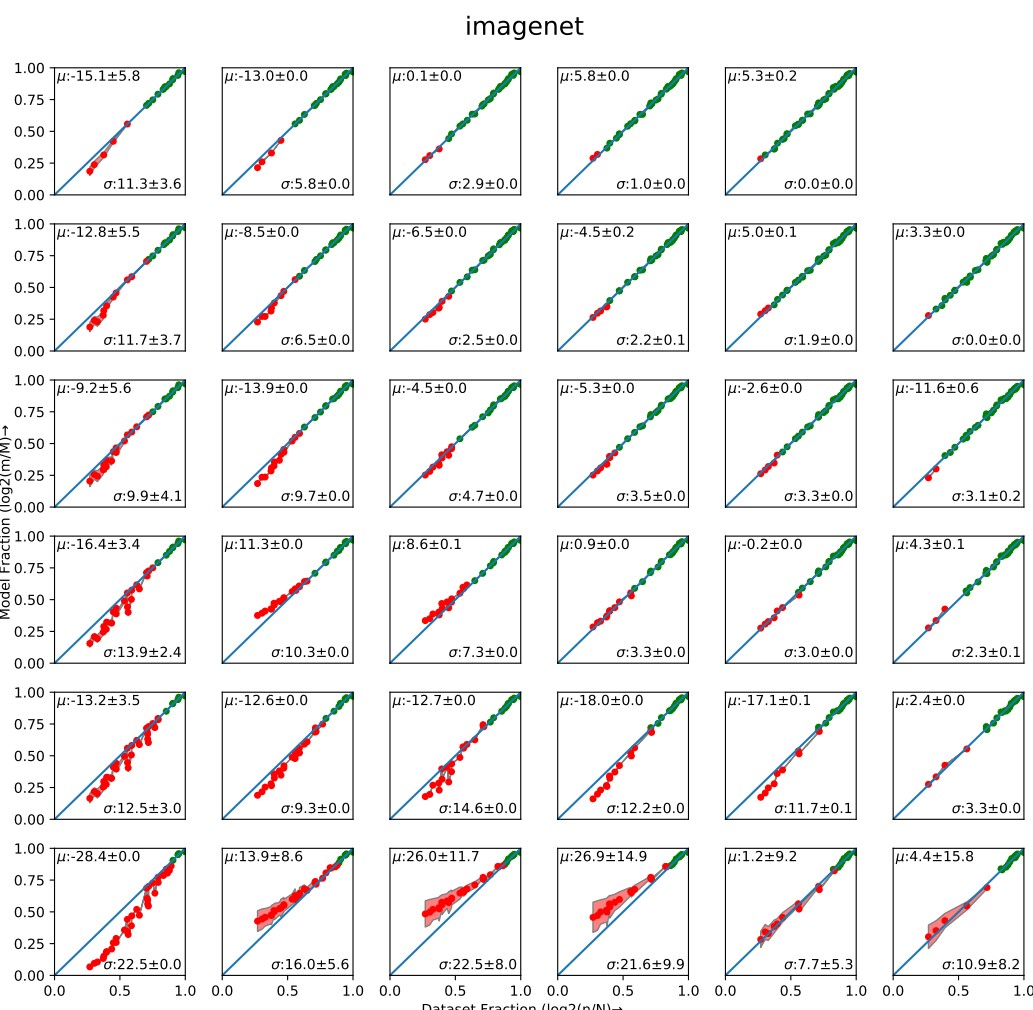

Figure 16: ImageNet extrapolation results.

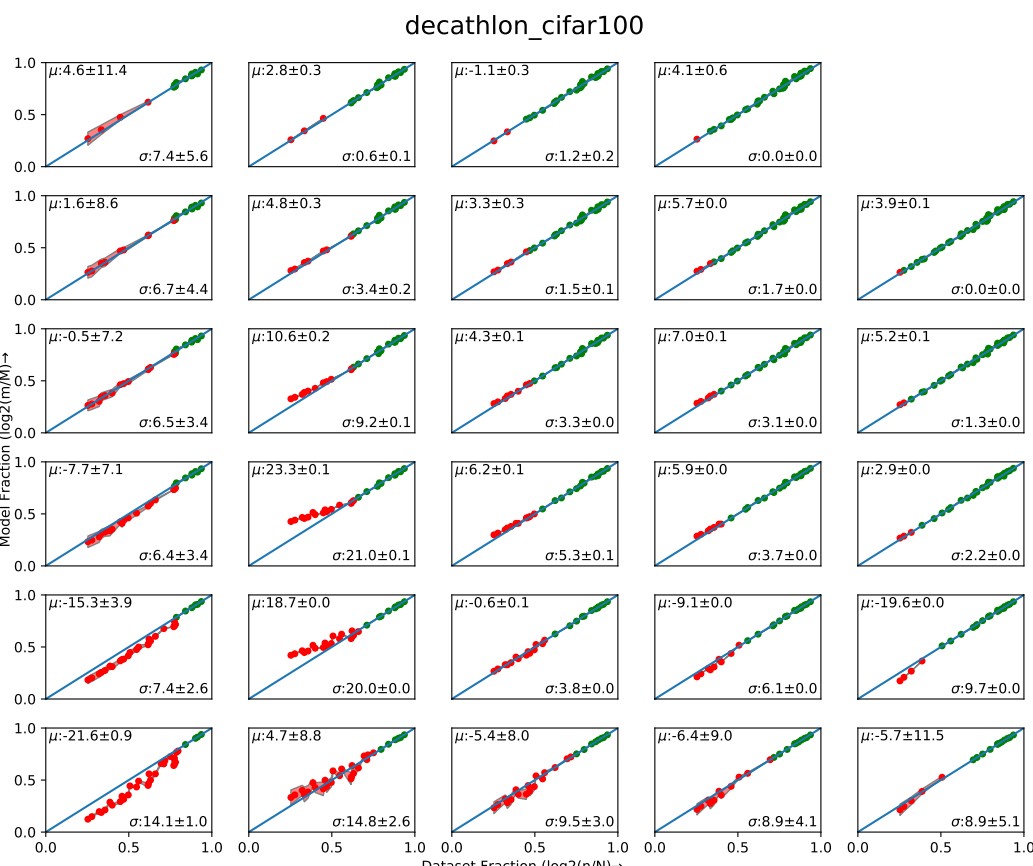

Figure 17: CIFAR100 Extrapolation Results

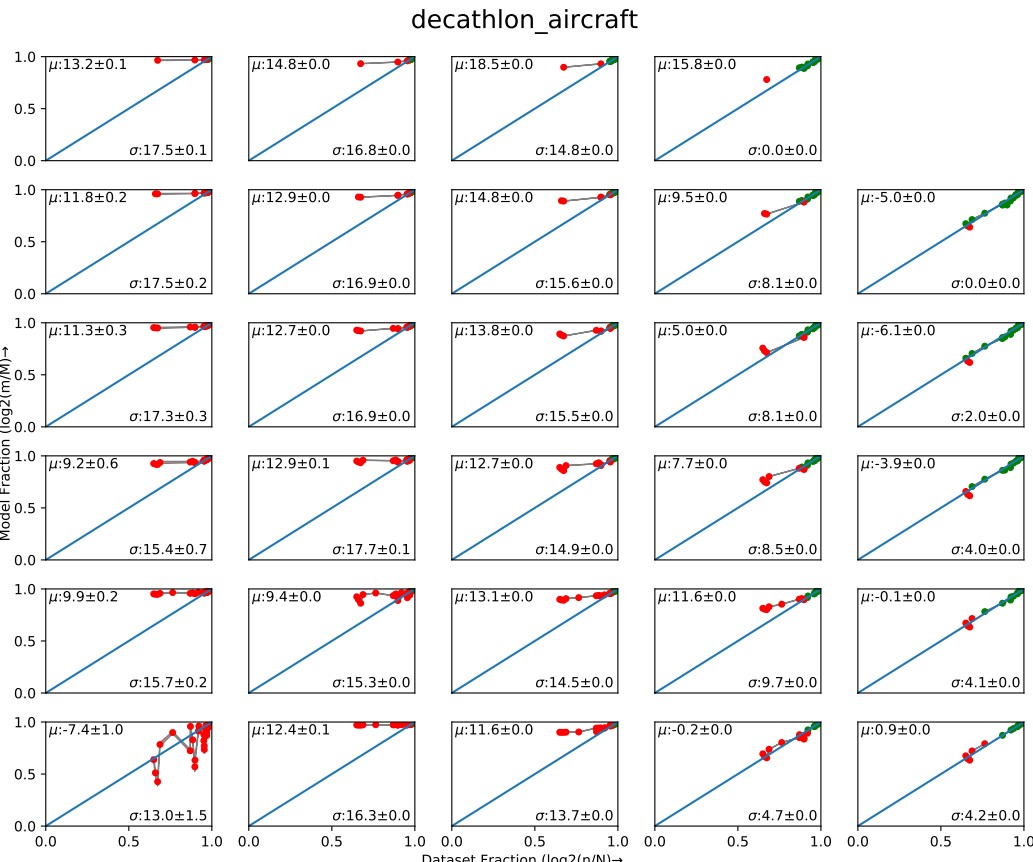

Figure 18: Aircraft extrapolation results.

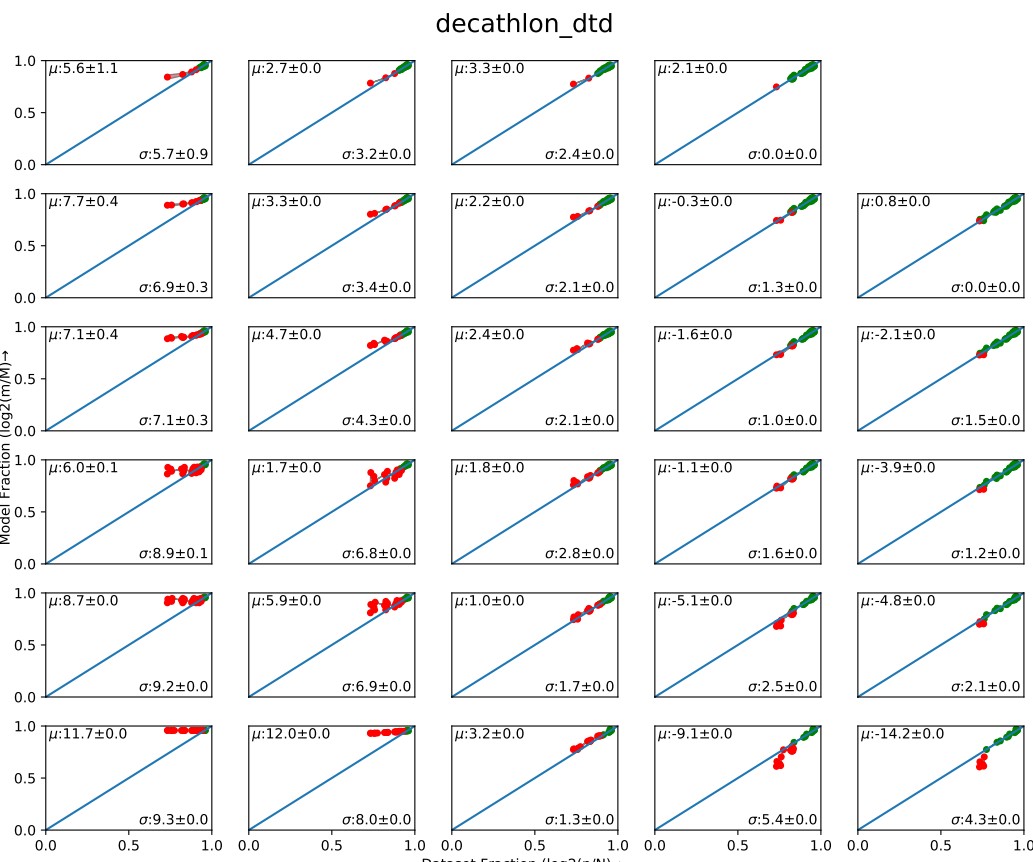

Figure 19: DTD Results

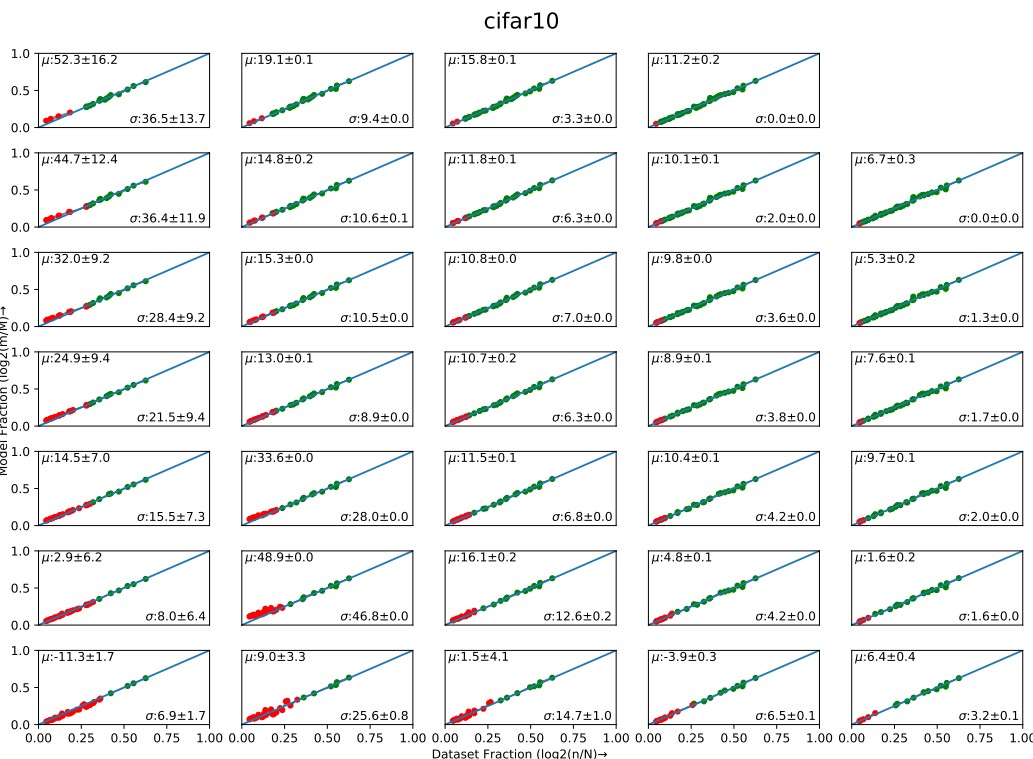

Figure 20: CIFAR10 extrapolation results.

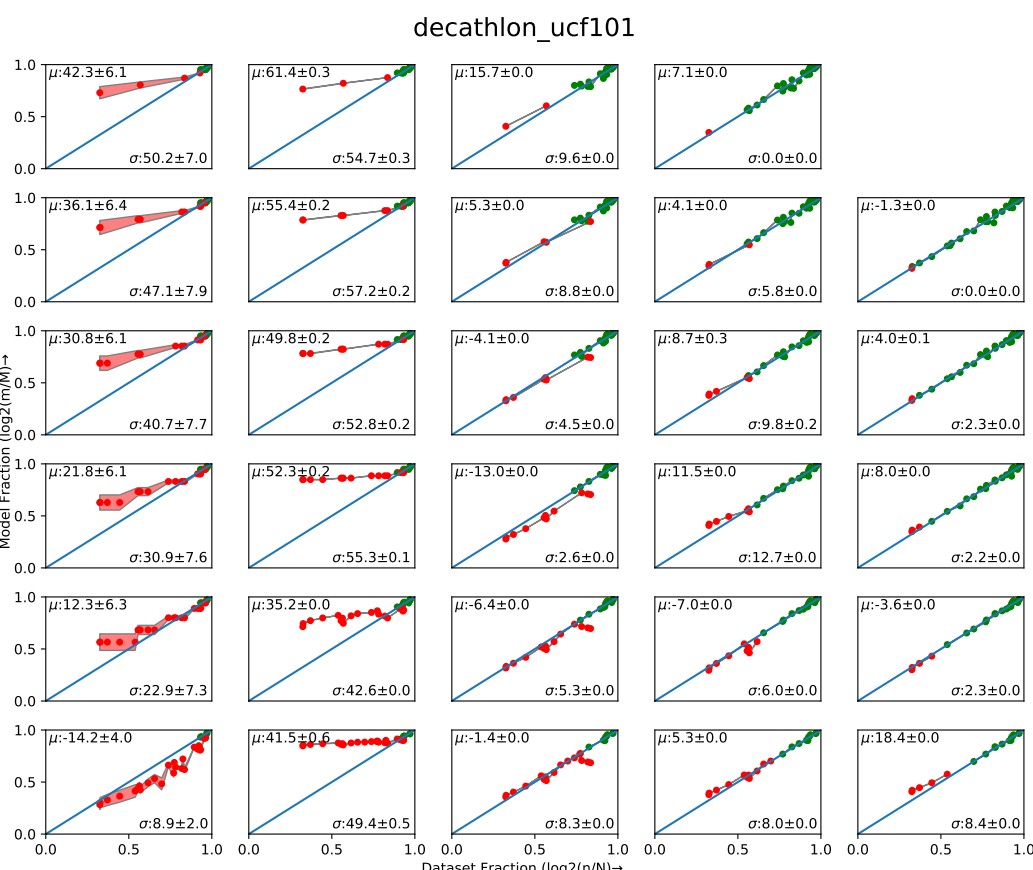

Figure 21: UCF101 extrapolation results.

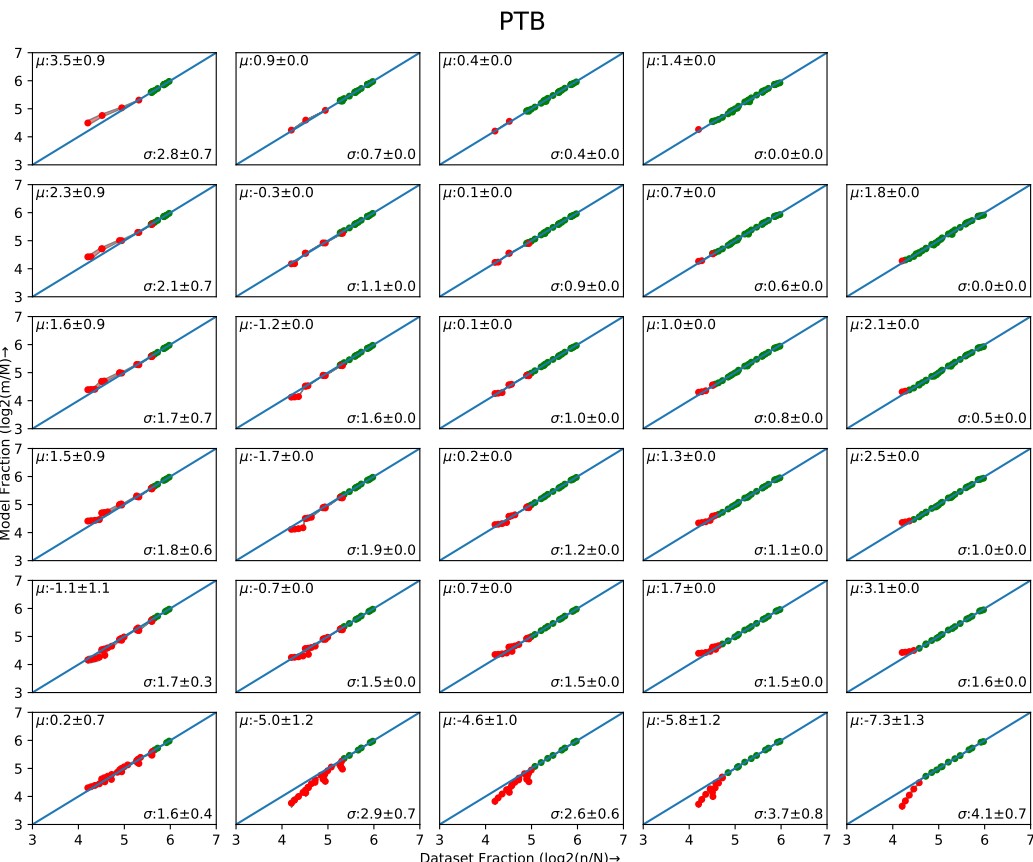

Figure 22: PTB extrapolation results.

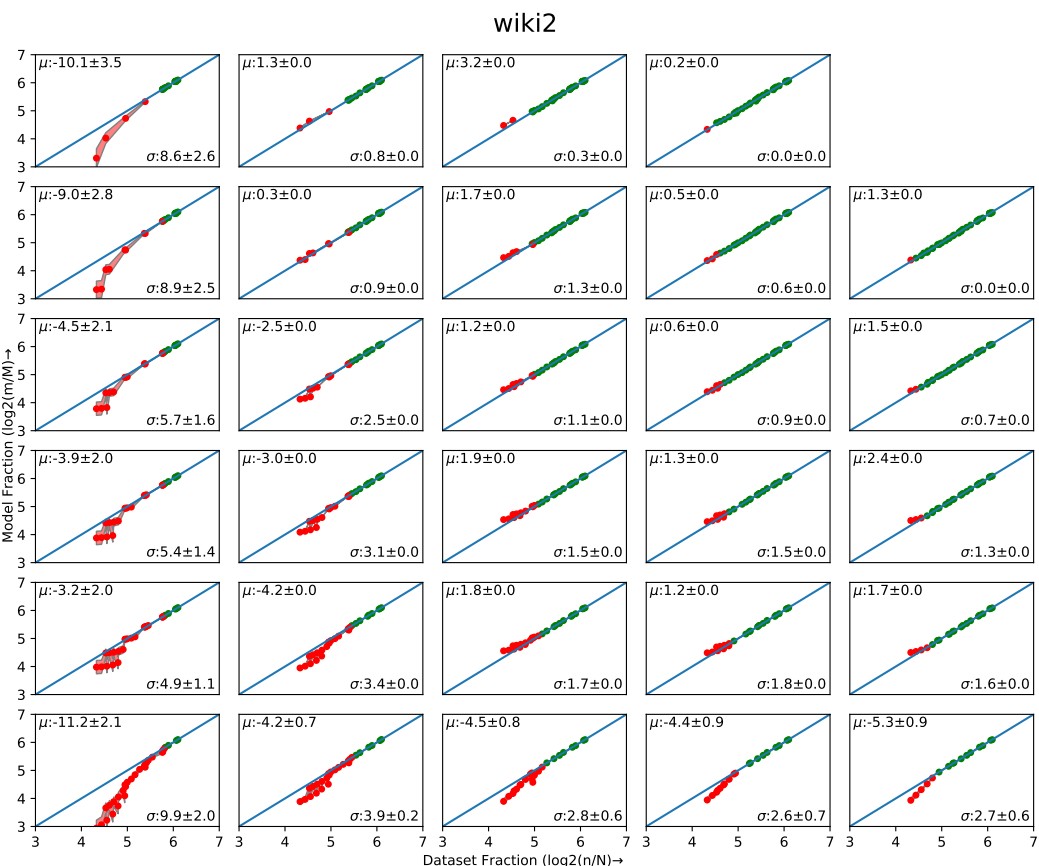

Figure 23: WikiText-2 extrapolation results.

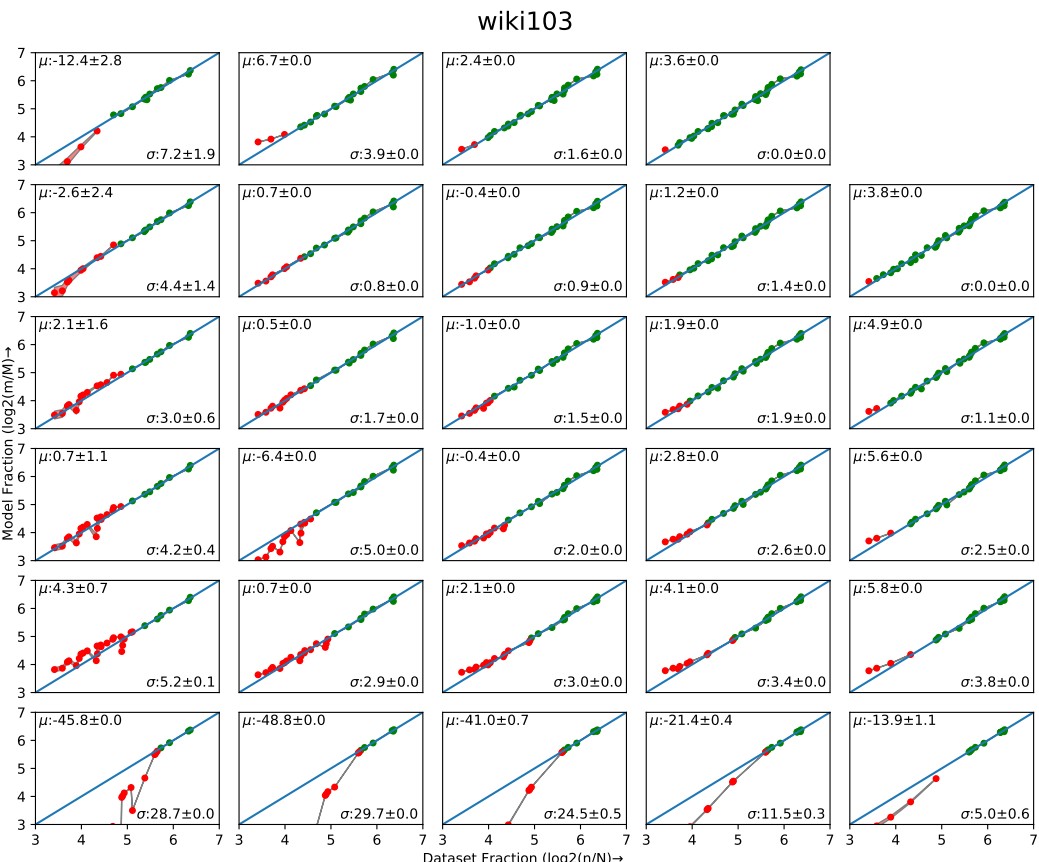

Figure 24: WikiText-103 extrapolation results.

