# OpenReview forum: "A Constructive Prediction of the Generalization Error Across Scales"
_ICLR.cc/2020/Conference — Accept (Poster)_

### Official Review · AnonReviewer3 · 2019-10-23
**Official Blind Review #3**

**Rating:** 8

**Review:**

This work proposes a functional form for the relationship between <dataset size, model size> and generalization error, and performs an empirical study to validate it. First, it states 5 criteria that such a functional form must take, and proposes one such functional form containing 6 free coefficients that satisfy all these criteria. It then performs a rigorous empirical study consisting of 6 image datasets and 3 text datasets, each with 2 distinct architectures defined at several model scales, and trained with different dataset sizes. This process produces 42-49 data points for each <dataset, architecture> pair, and the 6 coefficients of the proposed functional form are fit to those data points, with < 2% mean deviation in accuracy. It then studies how this functional form performs at extrapolation, and finds that it still performs pretty well, with ~4.5% mean deviation in accuracy, but with additional caveats.

Decision: Accept. This paper states 5 necessary criteria for any functional form for generalization error predictor that jointly considers dataset size and model size, then empirically verifies it with multiple datasets and architectures. These criteria are well justified, and can be used by others to narrow down the search for functions that approximate the generalization error of NNs without access to the true data distribution, which is a significant contribution. The empirical study is carefully done (e.g., taking care to subsample the dataset in a way that preserves the class distribution). I also liked that the paper is candid about its own limitations. A weakness that one might perceive is that the coefficients of the proposed functional form still needs to be fit to 40-ish trained NNs for every dataset and training hyperparameters, but I do not think this should be held against this work, because a generalization error predictor (let alone its functional form) that works for multiple datasets and architecture without training is difficult, and the paper does include several proposals for how this can still be used in practice.
(Caveat: the use of the envelope function described in equation 5 (page 6) is not something I am familiar with, but seems reasonable.)

Issues to address:
- Fitting 6 parameters to 42-49 data points raises concerns about overfitting. Consider doing cross validation over those 42-49 data points, and report the mean of deviations computed on the test folds. The extrapolation section did provide evidence that there probably isn't /that/ much overfitting, but cross validation would directly address this concern.
- In addition, the paper provides the standard deviation for the mean deviations over 100 fits of the function as the measure of its uncertainty, but I suspect that the optimizer converging to different coefficients at different runs isn't the main source of uncertainty. A bigger source of uncertainty is likely due to there being a limited amount of data to fit the coefficients to. Taking the standard deviation over the deviations measured on different folds of the data would be better measure of uncertainty.

Minor issues:
- Page 8: "differntiable methods for NAS." differentiable is misspelled.

**Experience Assessment:**

I have published one or two papers in this area.

**Review Assessment: Checking Correctness Of Derivations And Theory:**

I assessed the sensibility of the derivations and theory.

**Review Assessment: Checking Correctness Of Experiments:**

I carefully checked the experiments.

**Review Assessment: Thoroughness In Paper Reading:**

I read the paper thoroughly.

---

> ### Author Response · Authors · 2019-11-11
> **Response to Review #3**
>
> Thank you for your thorough and helpful review. We also believe that the criteria we identified will be useful for others in narrowing the search for functions that approximate the generalization error of NNs in realistic settings with no access to the true data distribution.
>
> Concerns regarding overfitting and uncertainty estimation: Given your suggestion, we performed 10-fold cross validation in all tasks and found high quality results and cross-fold consistency. We now report updated cross-val for all results in section 6 including figures 3,4 and in the newly-added figure 5. We believe that this addresses both the overfitting concern and the uncertainty estimation concern.
> Additional evidence that there is no overfitting is the good extrapolation results (section 7), as acknowledged by the reviewer.
>
> Regarding the envelope function (equation 5): This form of function is a simple case of the (complex) rational function family (simple pole at $\eta$, simple zero at the origin in this case). This family arises naturally in transitory systems in control theory and electrical engineering, e.g., when considering the frequency response of systems. It captures naturally powerlaw transitions. With that said, as we stress in the end of section 5, the particular choice of envelope is merely a convenience one and there may be other such functions / refinements. We leave further exploration of this aspect to future work.
>
> We have fixed the misspelling in “differentiable”. Thanks for pointing this out.

---

> > ### Comment · AnonReviewer3 · 2019-11-15
> > **Response to Response to Review #3**
> >
> > I would like to thank the authors for updating the results of their error estimation experiments with 10 fold cross validation, which addressed my overfitting and uncertainty estimation concerns. I wish them the best of luck.

---

### Official Review · AnonReviewer2 · 2019-10-23
**Official Blind Review #2**

**Rating:** 6

**Review:**

Summary:
This paper proposes a functional form to model the dependence of generalization error on a held-out test set on model and dataset size. The functional form is derived based on empirical observations of the generalizing error for various model and dataset sizes (sections O1, O2, and O3) and on certain necessary criteria (C1, C4 and C5). The parameters of the function are then fit using linear regression on observed data. The authors show that the regressed function \(\epsilon(m,n)\) is able to predict the generalization error for various \(m\) and \(n\) reasonably accurately.

Major Points:
- While the current experiments are a good start, I do not think they are extensive enough to count as strong evidence for the  power-law form of \(\epsilon(m,n)\). I would ideally like to see results on more optimizers, at the very least for Adam, even if for fixed hyper-parameters. As far as I understand, this involves only minor changes in the code since reasonable hyperparameters required for the convergence of Adam have been extensively studied. If the form still holds true then the results from this work can be more reliably used for small-scale network development and in making trade-off choices (as discussed in section 8).
- Given the current form of the paper, the abstract and introduction should be modified to reflect the fact that only limited architectures and optimizers were experimented with, and the claims of the paper are not experimentally validated in general.

Minor Points:

- It would be nice if more network architectures were analysed (such as VGG and DenseNets).
- It would be nice if different stopping criteria were analysed.
- It would greatly benefit the reader if eq. 5 were expanded.

Overall, I think this is a well written paper and provides good insight into the behaviour of the error landscape as a function of model and dataset size. The paper’s primary drawback is the restrictive setting under which the experiments are performed. Therefore, I am not convinced that the power-law form of the generalization error would hold when the experimental settings are marginally different (like when using the Adam optimizer or a VGG-like architecture). I think this work would have much greater impact if the authors can show that the power-law form holds for a larger variety of architectures and optimizers thus allowing researchers to more confidently incorporate the results of this work into the design and training deep neural networks.

Rebuttal Response
I would like to thank the authors for their response. The results of additional experiments as described in Section 6.2 and in Figure 5 do indeed provide stronger evidence of the power-law form of the error function. In light of this, I have changed my original rating.



**Experience Assessment:**

I do not know much about this area.

**Review Assessment: Checking Correctness Of Derivations And Theory:**

I assessed the sensibility of the derivations and theory.

**Review Assessment: Checking Correctness Of Experiments:**

I assessed the sensibility of the experiments.

**Review Assessment: Thoroughness In Paper Reading:**

I read the paper at least twice and used my best judgement in assessing the paper.

---

> ### Author Response · Authors · 2019-11-11
> **Response to Review #2**
>
> Thank you very much for your thoughtful review.
>
> We would like to point out that our experiments include multiple architectures (WRN and ResNet for image classification, LSTM and transformers for language modeling) and optimizers (SGD for image classification, SGD and Adam for language modeling). These were chosen according to standard implementations in the literature.
>
> However, we agree that it is important to demonstrate the results on a greater variety of architectures and optimizers and in particular in a manner that allows to assess the stability with respect to changing them for a specified task. Following your suggestion, we have therefore added experiments with both VGG and DenseNet, each trained with both SGD and Adam, on CIFAR100. The results conform with good agreement to the functional form defined in Eq. 5, with fit quality quantitatively very similar across all the architectures/optimizers settings in these experiments, and in particular reaching small divergences.  We added a new section (6.2) and figure (Fig. 5) for these experiments.
>
> We do believe that the variety of architectures/optimizers examined over a variety of tasks (extending to large datasets over both vision and language processing) in this study, augmented with the explicit additions following your valuable feedback, experimentally cover a meaningful chunk of settings, which supports our conclusions. We hope you will reevaluate the paper in light of these additions, and welcome any additional feedback.

---

### Official Review · AnonReviewer1 · 2019-10-25
**Official Blind Review #1**

**Rating:** 1

**Review:**

This paper explores the relation among the generalization error of neural networks and the model and data scales empirically. The topic is interesting, while I was expecting to learn more from the paper, instead of some well-known conclusions. If the paper could provide some guidance for model and data selection, that would be an interesting paper for the ICLR audience.  For instance, how deep should a model be for a classification or regression task? What is the minimum/maximum layers of a deep model? How much data is sufficient for a model to learn? What is the minimum/maximum size of the data set? Do we really need a large data set or just a subset that covers the data distribution? What's the relation between the size of a model and that of a data set? By increasing the depth/width of a neural network, how much new data should be collected for achieving a reasonable performance? How about the gain of the task performance?

**Experience Assessment:**

I have read many papers in this area.

**Review Assessment: Checking Correctness Of Derivations And Theory:**

I did not assess the derivations or theory.

**Review Assessment: Checking Correctness Of Experiments:**

I did not assess the experiments.

**Review Assessment: Thoroughness In Paper Reading:**

I read the paper at least twice and used my best judgement in assessing the paper.

---

> ### Author Response · Authors · 2019-11-11
> **Response to Review #1**
>
> Thank you for your review.
>
> We are a bit surprised since the paper provides answers to the exact questions you raised as missing. We are sorry you missed it, and we have cleaned up the presentation so it is hopefully now clear that we do answer these questions and more. The answers, as you pointed out, were much desired and not known before.
>
> Below are answers resultant from eq. 5 to the specific questions the referee raised, with some added definitions to make them concrete.
>
> 1. “how deep should a model be for a classification or regression task? “
>
> We show in section 6.1 that the dependency of the classification error on the number of layers is also well approximated by eq. 5 (recall $m$ scales linearly with depth).
>
> So, if we consider some target error $\epsilon_{target}$, we can solve eq. 5 for m or n given the other or for both, attaining the m,n contour for $\hat{\epsilon}(m,n) = \epsilon_{target}$.
>
>
> 2. “What is the minimum/maximum layers of a deep model? “
>
> For a fixed dataset size, model scaling eventually contributes marginally to error reduction and becomes negligible when $bm^{-\beta} \ll n_{lim}^{-\alpha}$ (Eq. 5).
>
> Define the relative contribution threshold $T$ as satisfying $ T = \frac{n^{-\alpha} }{ bm^{-\beta}}$. (For example, $T=10$.) Then the maximal useful model size meeting threshold $T$ is:
> $$     m_{max}(T) = \left(bT\right)^{1/\beta} n_{lim}^{\alpha/\beta}  $$
>
> As for minimal depth, here too let’s consider a definition as a working example: what is the minimum depth that could meet a certain error level $\epsilon_{target}$ (if data is not a limit).
> For example, when the target error is small relative to the “random guess error” $\epsilon_0$ (equivalently when $ n^{-\alpha} + bm^{-\beta} \ll \eta$), by solving eq. 5 for $m$ we have:
>
> $$ m_{min} = \left(\frac{b}{\frac{\epsilon_{target}}{\epsilon_0}\eta-c_\infty}\right)^{1/\beta} $$
>
> 3. “How much data is sufficient for a model to learn? What is the minimum/maximum size of the data set?”
>
> Similarly to the above:
> Minimum data needed for target error (if model size is not a limit):
> $$ n_{min} = \left(\frac{1}{\frac{\epsilon_{target}}{\epsilon_0}\eta-c_\infty}\right)^{1/\alpha} $$
>
> 4. Maximum useful data (in the marginal sense $T$ for a limited size model, as above):
> $$n_{max}(T) = \left(1/bT\right)^{1/\alpha} m_{lim}^{\beta/\alpha} $$
> In particular, note that there is also a minimal amount of data and model size needed for better-than-random-guess error level, characterized by the location of the pole $\eta$: $n^{-\alpha}+bm^{-\beta}< \eta$
>
> 5. “Do we really need a large data set or just a subset that covers the data distribution?”
>
> Via careful dataset sub-sampling (as noted by reviewer 3) we show that indeed more data *is* needed to improve performance (reduce error) while holding the class distribution fixed (in expectation), for a given architecture and scaling policy. For directly viewing the error manifolds decoupling the dependency on model and data size, see figure 1 and in appendix C.
>
> 6. “What's the relation between the size of a model and that of a data set? “
>
> The joint form in Eq. 5 captures the relation between data-size and model-size (and error) completely.
>
> 7. “By increasing the depth/width of a neural network, how much new data should be collected for achieving a reasonable performance?”
>
> For example, from Eq. 5, it is clear that a sweet-spot in terms of balancing the effect of the data/model sizes on limiting the error is $n^{-\alpha} \approx bm^{-\beta}$ .
>
> When considering this sweet spot for example, increasing depth/width/both such that the model size $m$ is increased by a factor $f$ to a new size is $m’ = mf$, the corresponding increase in data maintaining the sweet-spot is $n’ = nf^{\beta/\alpha}$
>
> 8. How about the gain of the task performance?”
>
> The effect on the performance is given by evaluating Eq.5 for the initial and scaled $m,n$.
>
> For example, in the powerlaw region ($c_\infty \ll n^{-\alpha} + bm^{-\beta} \ll \eta$):
> The effect on the performance is $\epsilon’ = \epsilon f^{-\beta}$

---

### Author Response · Authors · 2019-11-11
**Updated Revision**

We would like the thank the reviewers for their helpful comments. We have updated the paper accordingly. Please see detailed responses in the individual comments on each review.

---

### Decision · Program_Chairs · 2019-12-19

**Decision:**

Accept (Poster)

**Comment:**

The paper presents a very interesting idea for estimating the held-out error of deep models as a function of model and data set size. The authors intuit what the shape of the error should be, then they fit the parameters of a function of the desired shape and show that this has predictive power. I find this idea quite refreshing and the paper is well written with good experiments. Please make sure that the final version contains the cross-validation results provided during the rebuttal.